# Early Intervention in Ulcerative Colitis: Ready for Prime Time?

**DOI:** 10.3390/jcm9082646

**Published:** 2020-08-14

**Authors:** Virginia Solitano, Ferdinando D’Amico, Eirini Zacharopoulou, Laurent Peyrin-Biroulet, Silvio Danese

**Affiliations:** 1Department of Biomedical Sciences, Humanitas University, Pieve Emanuele, 20090 Milan, Italy; virginia.solitano@humanitas.it (V.S.); ferdinando.damico@humanitas.it (F.D.); 2Department of Gastroenterology and Inserm NGERE U1256, University Hospital of Nancy, University of Lorraine, 54500 Vandoeuvre-lès-Nancy, France; peyrinbiroulet@gmail.com; 3Department of Gastroenterology, IBD Center, Humanitas Clinical and Research Center, IRCCS, Rozzano, 20089 Milan, Italy; eirinizachar@gmail.com

**Keywords:** ulcerative colitis, early intervention, window of opportunity, top-down strategy, treat-to-target

## Abstract

Growing evidence shows that ulcerative colitis (UC) is a progressive disease similar to Crohn’s disease (CD). The UC-related burden is often underestimated by physicians and a standard step-up therapeutic approach is preferred. However, in many patients with UC the disease activity is not adequately controlled by current management, leading to poor long-term prognosis. Data from both randomized controlled trials and real-world studies support early intervention in CD in order to prevent disease progression and irreversible bowel damage. Similarly, an early disease intervention during the so-called “window of opportunity” could lead to better outcomes in UC. Here, we summarize the literature evidence on early intervention in patients with UC, highlighting strengths and limitations of this approach.

## 1. Introduction

Ulcerative colitis (UC) is a chronic, relapsing, and disabling inflammatory bowel disease (IBD), affecting the rectum and a variable extent of the colon [1]. Growing evidence indicates that UC should be considered a progressive disease exactly as Crohn’s disease (CD), having the same tendency to progress to irreversible structural damage [2]. Consequently, the treatment goals are constantly evolving in order to change the progressive course of IBD, and clinicians are recommended to follow a target-driven approach [3], aimed at achieving not only clinical remission but also objective outcomes, such as endoscopic and histological remission [4]. It has been reported that early intervention with biologics can slow disease progression and improve long-term outcomes in IBD, reducing irreversible damage [5]. This is attributable to the existence of a “window of opportunity” for intervention, before inflammation and eventual bowel damage become established [6]. An increasing volume of evidence from CD patients has shown that early initiation of anti-tumor necrosis factor (TNF) therapy, which is defined as a “top-down approach”, is associated with better outcomes compared to the “step-up strategy”, and might modify the disease course [5]. However, the management of most patients with UC still relies upon treatment intensification driven by clinical symptoms [7]. To date, the UC therapeutic algorithm implies a rapid “step-up approach” with steroids and thiopurines, while in subjects who fail these therapies, biologics (anti-TNF agents, vedolizumab, and ustekinumab) or small molecules are administered [8]. Although early intervention is an established approach in CD, data on the impact of early intervention in UC are limited [9]. Considering the inadequate control of the inflammation burden with current treatments [10], it is legitimate to speculate that early intervention could prevent disease progression in UC [11,12]. Based on the hypothesis that UC and CD have a similar inflammatory burden and progressive nature, we first reviewed literature regarding early intervention in patients with IBD. Subsequently, we focused on the current state-of-the art in early intervention in UC, highlighting strengths and limitations of this approach. Finally, we also provided insights on the importance of identifying risk factors to guide treatment decision according to a personalized management.

## 2. Ulcerative Colitis: A Progressive Disease

UC is considered by most physicians as a less progressive disease compared to CD [2]. However, similarly to CD, several studies have shown that uncontrolled UC is associated with structural damage and impaired gastrointestinal functioning [9]. This misconception explains the reluctant attitude to introducing more effective treatments earlier in the course of the disease [13]. Of note, proximal disease extension is detected in up to 54% of patients with UC during the course of disease [14]. Structural changes, such as colonic wall thickening and strictures, are reported in 1% to 11.2% of cases [15]. With regard to fibrosis, it is a well-known complication of CD, leading to clinically relevant strictures in almost one-third of patients in their disease course [15]. Although this complication has been neglected in UC for years, a significant degree of fibrosis and muscolaris mucosae thickening, which are linked to the inflammatory burden, have been recently reported [15].

In addition, anatomical changes such as rectal narrowing, and widening of the presacral space are described, supporting the concept that disease damage is a relevant issue in patients with UC [2,16]. Moreover, patients with UC with severe and persistent inflammation (10–15%) have an increased risk of developing colorectal cancer [17,18,19]. In long-term follow up studies, approximately 50% of patients reported clinical remission after an initial period of high disease activity, but the cumulative risk of relapse is 70–80% at 10 years, and almost half of patients need UC-related hospitalization [12]. Among those patients receiving corticosteroids (almost 50%), complete remission is achieved in about half of the subjects [12].

Furthermore, patients are often undertreated [20]. Significantly, approximately 10% of patients with UC undergo a colectomy after 10 years from diagnosis [21]. Although surgery represents an adequate and recommended strategy in patients refractory to medical treatment, postoperative complications should not be minimized [22]. A systematic review showed that early complications (e.g., infections, ileus, pouch related complications, and obstructions) occurred in up to 65% of patients after surgery, whereas late complications (e.g., pouchitis, fecal incontinence, and obstructions) occurred in up to 55% of cases [22]. In a European Federation of Crohn’s and Ulcerative Colitis Association (EFCCA) survey 42.7% of patients with UC reported a recurrence of symptoms after surgery, and over a quarter (*n* = 102/321, 31.8%) reported serious complications after surgery [23]. UC is also associated with substantial risk of progression to serious complications, patient morbidity, and poor quality of life [12,21]. Interestingly, a French study investigated the disability status of IBD patients through the Inflammatory Bowel Disease Disability Index (IBD-DI) questionnaire, finding no important difference between CD and UC groups [24].

## 3. Early Disease: The Definition

The first definition of early disease derives from the Paris consensus on early CD [25]. It was specifically proposed for disease-modification trials by an international group of IBD experts [25], and it included CD patients with a disease duration of ≤18 months from diagnosis and no prior or current treatment with disease-modifying drugs, such as immunomodulators (IM) and biologics. The need for an accepted definition of early disease comes from rheumatological studies, where the introduction of biological treatment at early stages of rheumatoid arthritis (RA) prevents joint damage and patient disability, modifying the disease course [26,27]. RA studies have demonstrated that early disease is highly receptive to treatments [28,29], suggesting that the “therapeutic window” could dramatically modify the natural history of disease [6]. The evidence of different inflammatory mechanisms in early versus late colitis in experimental murine models, and in humans, supported the current theory of considering early disease as a clinically distinct entity [30,31]. From a pathophysiological point of view, many cytokines are differentially expressed in different phases, such as IL-12 and IL-17 during early and late stages, respectively [31,32]. Even though in CD a definition of early disease has been established (≤18 months from diagnosis), in UC it is yet to be clearly defined [25].

## 4. Lessons from Early Intervention in CD

Data from both randomized controlled trials (RCTs) and observational cohort studies showed that early intervention with biological agents could slow CD progression, leading to improved long-term outcomes [5]. A post hoc analysis of the phase III CHARM and ADHERE trials [33] reported that adalimumab-treated patients with shorter disease duration (<2 years) were more likely to be in clinical remission at week 56 compared with longer disease duration subgroups (OR 0.97, *p* = 0.046). Moreover, three-year remission rates were higher in patients with <2 years disease duration compared to those with longer disease duration (>40% vs. ≃30%, respectively) [33]. A RCT conducted by D’Haens et al. [34] recruited newly diagnosed CD patients (average disease duration of 6 months), comparing the effectiveness of combined immunosuppression, with infliximab and azathioprine (top-down approach), with initial treatment with corticosteroids (step-up therapy). Patients in the former group achieved higher remission rates than the latter group at week 26 (60% and 35.9%, respectively; *p* = 0.0062) and 52 (61.5% and 42.2%, respectively; *p* = 0.0278), suggesting that introduction of a more intensive early treatment led to better outcomes [34]. Similarly, in a post hoc analysis of the SONIC trial, Colombel et al. [35] stratified patients according to disease duration (defined by the Paris consensus), revealing that early CD (≤18 months) was associated with higher rates of clinical remission (81% vs. 80%, *p* = 0.036) and other composite outcomes (such as clinical remission plus mucosal healing; 63% vs. 53.3%, *p* = 0.004), compared with non-early CD patients. A recent pooled analysis of data from several phase III clinical trials was in line with these findings [36].

In the real-life setting, early use of anti-TNF agents and/or IM, defined as treatment within a two-year period from diagnosis, was associated with a reduced risk of bowel strictures (hazard ratios (HR) 0.496, *p* = 0.004 for IM; HR 0.276, *p* = 0.018 for anti-TNF) compared to late initiation (after >2 years from diagnosis) of the treatment [37]. Risks of intestinal surgery (HR 0.322, *p* = 0.005), perianal surgery (HR 0.361, *p* = 0.042), and any disease complication (HR 0.567, *p* = 0.006) were reduced in early initiators of IM [37]. More recently, a Swiss group assessed the long-term outcomes (up to 10-year follow-up) of patients who received early (<24 months after diagnosis) or late (>24 months) anti-TNF treatment or were never treated with this drug class, and found that early anti-TNF administration was associated with a reduced risk of developing bowel stenosis (log-rank test: *p* < 0.001) [38]. Another large cohort study explored the impact of early (within the first two years of disease) treatment with anti-TNF agents on the rate of surgical resection, and reported a decreased rate of surgery and clinical secondary loss of response in the early anti-TNF cohort group (5.7% versus 30.7% (*p* < 0.001) and 45.3% vs. 67.2% (*p* = 0.006), respectively) [39]. A recent systematic review, with meta-analysis enrolling 18,471 patients from prospective clinical trials and real-world settings, confirmed these findings [40]. Finally, interesting data have come from a recent follow-up analysis of the CALM study [41]. In this RCT [42], CD patients with early disease (≤2 years), were randomly assigned to tight control (based on clinical symptoms combined with biomarkers) or conventional symptom-driven management groups, showing better clinical and endoscopic outcomes in the former arm. Patients who were in deep remission (defined as Crohn’s disease activity index (CDAI) <150, Crohn’s disease endoscopic index of severity (CDEIS) <4 with no deep ulcerations, and no steroids for ≥8 weeks) had a lower risk of major adverse outcomes over a median of 3 years (adjusted HR (aHR), 0.19; 95% CI, 0.07–0.31), highlighting the impact of early deep remission on long-term disease modification [41].

## 5. Evidence on Early Intervention in UC

Data on the impact of early intervention in UC are limited, and whether more intensive treatments prevent structural and functional complications is still debated [43]. Several observational studies have suggested that disease duration is not associated with therapy efficacy in UC [44,45,46,47,48], as summarized in Table 1. A cohort study conducted by Ma et al. [44] defined early intervention in UC as initiating anti-TNF treatment within three years of diagnosis; they assessed retrospectively the effect of early treatment on rate of colectomy and other UC-related complications in 115 patients. Authors found similar rates of colectomy (6 colectomies for 100 patient-years (PYs) of treatment vs. 2.7 colectomies per 100 PYs of treatment, *p* = 0.13), secondary loss response (49.1% vs. 58.6% versus *p* = 0.31), and hospitalization (43.9% vs. 27.6%, *p* = 0.07) between early and late initiators [44]. A multicenter study by Oussalah et al. [45] investigated predictors of short and long-term outcomes of infliximab in 191 patients with UC. They found that disease duration ≤50 months was an independent predictor of UC-related hospitalization (HR = 2.14, *p* = 0.006) [45]. On the other hand, early UC was not associated with improved long-term outcomes in a Canadian retrospective cohort including 213 patients [46]. Paradoxically, long disease duration was associated with higher one-year steroid free remission (adjusted odds ratio (aOR = 2.1; 95% CI, 1.2–3.5, per 10-year increase in disease duration, *p* = 0.061) and lower risks of infliximab failure (aHR = 0.59; 95% CI, 0.40–0.87, per 10-year increase, *p* = 0.0198) and colectomy (aHR = 0.49; 95% CI, 0.28–0.85, per 10-year increase, *p* = 0.0048) [46]. In agreement with these results, Mandel and colleagues did not reveal a benefit of early anti-TNF exposure (within 3 years from diagnosis) in patients with UC. Conversely, the need for hospitalization decreased by 40% (OR = 0.60, 95% CI 0.48–0.75, *p* < 0.001) in early-treated CD patients [48]. Shifting attention from anti-TNF to anti-integrins, no correlation was found between early disease duration (≤2 years) in UC and clinical remission, steroid-free remission, and endoscopic remission after six months of treatment with vedolizumab [47].

## 6. Outlook

Modifying the natural course of disease is a clear goal for UC management [49]. In addition, a new and more ambitious concept of “disease clearance” is emerging as a potential target in the treatment of UC [50]. This term includes symptomatic remission based on patient reported outcomes (PROs), together with mucosal healing, which encompasses both endoscopic and histological remission [50,51]. The early introduction of a biological therapy could allow the achievement of these targets, as clearly shown in patients with CD [40]. However, several issues persist regarding the role of early treatment in patients with UC.

First, available literature studies have shown that there is no relevant benefit for patients with UC treated with early biological therapy. It should be noted that a widely accepted definition of early disease in UC is lacking, leading to uncertainty over the exact meaning of this term and preventing definitive conclusions about the efficacy of this strategy from being drawn [25].

Second, most UC studies on this topic have a retrospective study design which is limited by indication bias, occurring when both treatment and outcome are influenced by a third factor, such as patients’ prognosis [9]. To overcome this limitation, large specifically designed randomized trials are required to assess whether early therapy initiation can positively influence disease outcomes of patients with UC, after comparing this with the standard approach. Of note, the impact of disease modification in UC may require an extended period, so an adequate follow-up (>3 years) is necessary to evaluate disease progression and long-term negative outcomes (e.g., fibrosis, colorectal cancer, colectomy) (Figure 1) [52].

An ongoing randomized multicenter study, called the SPRINT study (EudraCT number: 2020-003420-16), will compare the efficacy of top-down and step-up approaches in patients with UC, with a disease duration of less than two years. All patients will initially undergo oral corticosteroid treatment. After endoscopic reassessment at week 16, patients with endoscopic (Mayo score > 1) or histological (Nancy score > 1) disease will undergo dose escalation of therapy. In the top-down group, treatment with adalimumab will be started immediately. On the other hand, in the step-up group, patients will be treated first with steroids and immunosuppressants, while adalimumab will be started in case of non-response during subsequent visits (weeks 28 and 40). The study will last one year and will have a follow-up of three years during which the hospitalization and malignancy rates will be recorded, providing relevant information on early disease intervention in UC. Results from similarly designed trials will further clarify the long-term implications of this approach, given the paramount importance of improving the patient’s quality of life by reducing disability, need for surgery and hospitalization.

To date, it is not known how to decide whether a patient is eligible for early therapy or not. In the absence of clear data, the decision of an early approach should be individualized and based on the risk–benefit ratio of each specific UC patient [49,53]. In this new scenario, it is important to balance the potential benefits deriving from changing the natural UC course with the risk of experiencing undesirable adverse events or overtreatment [49]. The long-term cumulative risks of immunosuppressive drugs (e.g., opportunistic infections and malignancies) should not outweigh the potential advantages of starting biologic therapy early, preventing the development of irreversible bowel injury [54]. Gathering all the information on the risk and benefit parameters has been recently proposed as the standard of quality when opting for the best individual treatment strategy [54].

Therefore, as already demonstrated in RA and CD, identification and selection of the best patient population who may benefit from early intervention are crucial [55,56,57]. Several prognostic factors of complicated UC have been identified and should be considered in the decision making process: young age at diagnosis, extensive colitis, presence of primary sclerosing cholangitis, need for early corticosteroids, and endoscopic disease severity [58,59,60,61]. It is likely that patients with these characteristics have a greater risk of developing a disease with a severe course and that early treatment may be recommended in these categories of patients [62].

Another potential obstacle to the use of an early strategy in UC is the lack of familiarity with new goals by many clinicians, who continue to manage patients with UC symptomatically. The concept of a treat-to-target’ strategy, adapted from RA, emerged in 2015 with the aim of altering the natural course of IBD [3]. The selecting therapeutic targets in inflammatory bowel disease (STRIDE) committee proposed the combination of symptomatic remission and endoscopic healing (e.g., Mayo endoscopic subscore of 0 in UC) as feasible therapeutic goals [3]. Hence, adjusting therapy according to the achievement (or not) of predetermined targets is expected to be an integral part of daily practice. Knowledge and awareness of the new therapeutic goals appear of critical importance for the management of IBD patients. An Australian survey, including 61 gastroenterologists, revealed that over a third of respondents were unfamiliar with the concept of treat-to-target and did not use it in daily clinical practice [7]. In line with these results, a Swiss survey reported that most gastroenterologists based their therapeutic decisions on clinical activity (70%), while endoscopic activity of disease and biomarkers were considered in a small percentage of cases (24% and 6%, respectively) [63]. For this reason, standardization of patient management is increasingly desirable and should be based on solid scientific evidence rather than on individual hospital recommendations in order to guarantee a minimum standard of care and to provide the best approach for patients.

In the meantime, we encourage physicians to raise the bar in terms of therapeutic goals, aiming at “disease clearance” as an ultimate achievable objective (Figure 2).

## 7. Conclusions

UC is a disabling disease that can potentially lead to the development of progressive damage. The available evidence does not support the early biologic intervention in patients with UC, in contrast with data on RA and CD. However, the lack of a validated definition of early UC and the low level of evidence from retrospective studies prevent definitive conclusions. Large randomized prospective studies are needed to assess the efficacy of this strategy and to identify predictors of complicated disease in order to select the optimal candidates for early intervention. Finally, clinicians’ awareness of changing treatment paradigms is fundamental, and efforts should be made to implement the standardization of patient management and to avoid irreversible bowel injury.

## Figures and Tables

**Figure 1 jcm-09-02646-f001:**
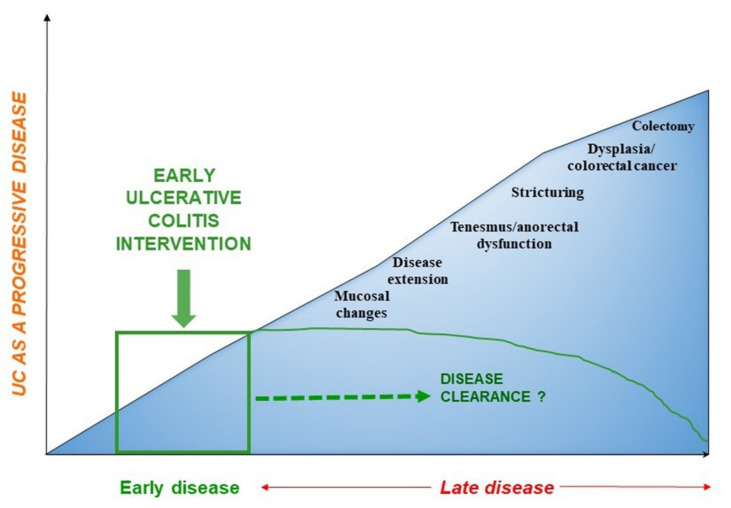
Disease progression in patients with ulcerative colitis: towards “disease clearance”.

**Figure 2 jcm-09-02646-f002:**
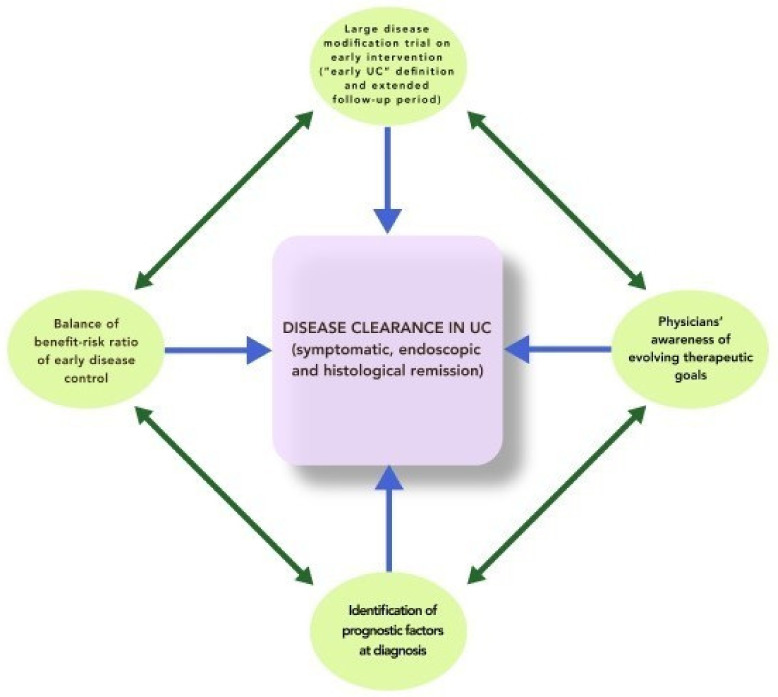
Combining strategies towards “disease clearance”.

**Table 1 jcm-09-02646-t001:** List of relevant cohort studies on early treatment in UC.

Author (Year)	Ref	Study Design	Study Population (Number)	Definition of Early UC	Outcomes	Significant Results
Ma C., et al. (2016)	[44]	Retrospective Observational	UC (115)	Within 3 years of diagnosis	Colectomy UC-related hospitalization Secondary loss of response	Early initiators vs. late initiators 6 for 100 PY vs. 2.7 per 100 PY, *p* = 0.13 43.9% vs. 27.6%, *p* = 0.7 49.1 vs. 58.6%, *p* = 0.31
Oussalah A., et al. (2010)	[45]	Retrospective Observational	UC (191)	≤50 months	Predictors of short- and long-term outcomes of IFX	Short duration at IFX initiation predicts hospitalization: HR = 2.14, 95% CI = 1.25–3.66, *p* = 0.006
Murthy S.K., et al. (2015)	[46]	Retrospective Observational	UC (213)	Not available	Annual SFR IFX failure with discontinuation colectomy	Longer disease duration aOR = 2.1; 95% CI, 1.2–3.5, *p* = 0.061 aHR = 0.59; 95% CI, 0.40–0.87, *p* = 0.0198 aHR = 0.49; 95% CI, 0.28–0.85, *p* = 0.0048
Faleck D.M., et al. (2019)	[47]	Retrospective Observational	IBD (1087) [CD (650)/UC (417)]	≤2 years	Clinical remission rates CSFR Endoscopic remission within 6 months with VDZ	Early-stage vs. late-stage CD aHR = 1.59; 95% CI, 1.02–2.49, *p* < 0.2 aHR = 3.39; 95% CI, 1.66–6.9, *p* < 0.2 aHR = 1.90; 95% CI, 1.06–3.39, *p* < 0.2
Mandel M.D., et al. (2014)	[48]	Retrospective Observational	IBD (194) [CD (152)/UC (42)]	Within 3 years from diagnosis	Hospitalization rates and predictors during anti-TNF therapy	Early anti-TNF exposure CD OR = 0.60, 95% CI 0.48–0.75, *p* < 0.001

aHR, adjusted hazard ratios; aOR, adjusted odd ratios; CD, Crohn’s disease; CSFR, corticosteroid-free remission; IBD, inflammatory bowel diseases; IFX, infliximab; PY, patient-years; SFR, steroid-free remission; TNF, tumor necrosis factor; UC, ulcerative colitis; VDZ, vedolizumab.

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
