# Peer review of "Early Intervention in Ulcerative Colitis: Ready for Prime Time?"

_jcm, 2020, doi:10.3390/jcm9082646_

Round 1

Reviewer 1 Report

Whether or not early intervention in UC is ready for prime time, there is no doubt that the time is ripe for a thorough review of the topic. This study provides exactly the needed survey. It is comprehensive and well documented.

I have a few comments:

  1. Since the authors stipulate in their very first paragraph that early-intervention is well established in CD, it does not seem necessary to devote all of page 3 (lines 90-142) to restating this entire argument.
  2. It might be preferable to change all allusions to “early UC” to “early-intervention UC.”
  3. Although it may seem obvious to the authors that the retrospective design of previous studies is flawed (lines 181-182), it could be useful to readers to point out that the fatal defect in all these studies is the inevitably of indication bias.
  4. Regarding Figure 1, we might ask for the specific evidence supporting “fibrosis” as a stage in UC disease progression. Stricturing is well recognized, but most studies suggest that the pathogenesis of strictures in UC involves muscular hypertrophy at least as much fibrosis per se. Perhaps “stricturing” or “myofibrosis” might be a more inclusive term.

Finally, while not the principal focus of this review, a closely related question is whether or not early intervention reduces the incidence of late proximal spread of distal colitis to involve the whole colon. This phenomenon is said to occur in 10-30% of left-sided colitis cases, and it is almost inevitably catastrophic in severity. Perhaps food for a separate study?

Reviewer 2 Report

This is a valuable area to review but I feel the article is a little thin on substance and lacks balance. The problem with ulcerative colitis is that the majority of patients do not develop disabling disease or long-term damage. In fact mortality rates associated with UC are little different from the general population and it is clear older estimates of CRC risk were exaggerated.

A substantial part of the review is devoted to the disease modifying effects of early aggressive treatment in Crohn’s disease and rheumatoid arthritis which could be greatly edited or referred to in one or two sentences.

More of a consideration of the differences in biology between UC and CD would be useful, in particular the differing propensities to produce fibrosis and the much greater tendency for UC to go into complete remission following treatment with corticosteroids and for it to have much more of a pattern of relapse and complete remission. In ulcerative colitis symptoms are a much better guide to what is going on at the tissue level than in CD. The authors present the evidence that in limited retrospective studies that unlike in CD the effectiveness of treatment with biologics is not impacted by the duration of disease.

In older text books the standard mantra used to be that the first episode of ulcerative colitis was the worst. This often seems to be the case from clinical experience, but I cannot find any evidence in the literature to directly support this.

A crucial paragraph begins on line 189 and needs breaking up into smaller paragraphs as a number of ideas are introduced: firstly there is the proposed trial; secondly the risk benefit of early intervention; thirdly prognostic factors for the development of continuously active severe  UC; fourthly clinician familiarity with treatment goals. All of these topics could be expanded at the expense of a very long paragraph on the benefits of early intervention in Crohn’s.

The first line of the conclusion needs toning down- in most cases UC is not a disabling disease leading to progressive damage. Rather than calling for large randomized controlled trials in the conclusion, perhaps the emphasis should be on better understanding of the natural history and which groups are at risk of developing complications.

Round 2

Reviewer 2 Report

I am now happy that the authors have addressed my concerns and that the manuscript is suitable for publication